# Exploring Perceived Stress among Students in Turkey during the COVID-19 Pandemic

**DOI:** 10.3390/ijerph17238961

**Published:** 2020-12-02

**Authors:** Imran Aslan, Dominika Ochnik, Orhan Çınar

**Affiliations:** 1Faculty of Health Sciences, Bingöl University, 12000 Bingöl, Turkey; 2Faculty of Medicine, University of Technology, 40-555 Katowice, Poland; dominika.ochnik@wst.pl; 3Faculty of Economics and Administrative Sciences, Ağrı İbrahim Çeçen University, 04000 Ağrı, Turkey; orhanar@gmail.com; 4Atatürk University, 2500 Erzurum, Turkey

**Keywords:** COVID-19, Satisfaction with Life Scale (SWLS), Perceived Stress Scale (PSS), Generalized Anxiety Disorder (GAD) scale, depression

## Abstract

Students have been highly vulnerable to mental health issues during the COVID-19 pandemic, and researchers have shown that perceived stress and mental health problems have increased during the pandemic. The aim of this study was to reveal the prevalence of perceived stress and mental health among students during the pandemic and to explore predictors of stress levels. A cross-sectional study was conducted on a sample of 358 undergraduates from 14 universities in Turkey, including 200 female students (56%). The measurements used in the study were the Generalized Anxiety Disorder 7-item (GAD-7) scale, Patient Health Questionnaire (PHQ-8), Satisfaction with Life Scale (SWLS), Perception of COVID Impact on Student Well-Being (CI), Perceived Stress Scale (PSS-10), Physical Activity Scale (PA), and a sociodemographic survey. Students reported high perceived stress, mild generalized anxiety, and low satisfaction with life. More than half of the students met the diagnostic criteria of GAD (52%) and depression (63%). Female and physically inactive students had higher PSS-10 levels. A hierarchical linear regression model showed that after controlling for gender and negative CI, anxiety and physical inactivity significantly predicted high perceived stress. The study shows that students’ mental health during the pandemic is at high risk.

## 1. Introduction

The coronavirus disease (COVID-19) pandemic is a global experience that is unique in modern world history. Since it affects widely disparate areas of life on the social and individual levels, it is classified as an existential experience [1]. The ongoing pandemic has enforced social isolation, which is strongly related to psychological distress and high anxiety and stress levels [2,3,4,5,6]. There are particular social groups who are exposed to the SARS-CoV-2 virus and vulnerable to deteriorated mental health. The most exposed groups are elderly people, medical personnel, and international migrant workers [7,8]. Although young adults are the least exposed [9], they are the most vulnerable group with regard to the deterioration of mental health [10,11]. It is worth noting that high stress and anxiety levels were observed in students before the pandemic as well [12,13]. In addition, this group is considered to be at particular risk of high stress levels [14]. Therefore, students are highly vulnerable with regard to the negative psychological consequences of the COVID-19 pandemic, such as high stress levels. Hence, perceived stress levels and mental health of students during the pandemic require monitoring and in-depth research.

### Perceived Stress and Mental Health during the COVID-19 Pandemic

Psychological stress emerges from an imbalance between an individual’s perception and external demands. Perceived stress refers to the assessment of the degree to which the situation in one’s life is seen as stressful; therefore, it is related to the subjective assessment of life events [15]. It is related to assessing how unpredictable, uncontrollable, and overloaded individuals find their lives [16]. Mental health is analyzed in this study through the Generalized Anxiety Disorder (GAD) scale and depression symptoms according to the Diagnostic and Statistical Manual of Mental Disorders, fourth edition (DSM-IV) criteria. Based on those criteria, GAD is characterized by persistent and excessive worry about a number of different things. It relates to anxiety as a state [17]. Depression is one of the most common yet treatable mental health disorders [18]. It refers to a number of symptoms including depressed mood, loss of interest in most or all activities, loss of energy, or feeling of worthlessness [19]. Perceived stress is strongly related to both anxiety [20] and depression symptoms [21]. The relationship between perceived stress and mental health is complex, and the direct cause-and-effect relationship is not clear [21].

Public mental health during COVID-19 deserves special attention; based on previous experiences, pandemic outbreaks are related to generalized fear that can impede infection control and lead to depression, anxiety, and post-traumatic stress [22]. Previous health emergencies have shown psychological consequences of quarantine [23], such as high stress levels [24] and depression [25], especially in younger women [26]. Therefore, even though lockdown can decrease the spread of COVID-19 [27], it can deteriorate mental health. A survey carried out during the self-isolation, lockdown, and social distancing periods revealed that female gender, younger age, lower annual income, smoking, and multiple physical morbidities were found to be correlated with a higher risk of mental health problems [28] and lower satisfaction with life [27,29,30].

A national study among Chinese people showed that 35% of participants experienced psychological distress during the COVID-19 pandemic. Psychological distress was higher in women, people between 18 and 30 years or over 60 years of age, and those with higher education [31]. Higher psychological distress in the younger population and women was noticed also in an Italian study (higher perceived stress, anxiety, and depression) [32] and in other Chinese research (higher stress and anxiety). Research on a general Chinese population showed a prevalence of moderate to severe depressive and anxiety symptoms of 16.5% and 28.8%, respectively [30]. Among people who experienced self-isolation for 14 days, anxiety was positively correlated with stress [33]. 

Students are particularly vulnerable to mental health problems, as a study before the COVID-19 pandemic showed [34]. Among variables related to mental health were older age and female sex, although the relationship was relatively modest. Therefore, it was assumed that mental health disorder symptoms were widely distributed in the general student population. The prevalence of depression and anxiety was 29% and 21%, respectively, among medical students [12]. Among Turkish students, 43.5% presented moderate to high anxiety, 18.6% moderate to high depression, and 30.9% moderate to high stress levels [35].The prevalence of moderate and severe symptoms of depression, anxiety, and stress was respectively 29.5%, 50.3%, and 39.9% among final-year Turkish students before the pandemic [36]. Turkish female students had higher depression, anxiety, and stress levels compared to male students [37].

The impact of the COVID-19 pandemic on mental health and education is expected to be highly noticeable [38]. A comparison of students’ mental health before and during the pandemic showed significant deterioration in aspects of depression, anxiety, and perceived stress, with female students being more vulnerable to decreased mental health compared to male students [11]. Most of the students (71%) indicated increased perceived stress and anxiety due to the pandemic [39]. Additionally, a Turkish study showed that students experienced higher levels of anxiety, depression, and perceived stress during the COVID-19 pandemic compared to the pre-pandemic period [40].

The prevalence of stress during the COVID-19 outbreak among first-year French students was 61.6% [41], while in Saudi Arabia 30.2% of students experienced high perceived stress, with more female students reporting high perceived stress levels than male students (34% vs. 19%) [42]. The prevalence of moderate and severe anxiety symptoms during the pandemic was nearly 4% in Chinese students [43], whereas in Polish university students it was 35% [29]. Living in an urban area, having family income stability, and living with parents were related to lower anxiety, whereas having relatives or acquaintances infected with COVID-19 was related to higher anxiety [44]. However, research among students in Bangladesh showed that living with parents and in an urban area was linked to higher depression and anxiety. This research also revealed different patterns of gender in mental health during the pandemic, as male students were characterized by higher depression and anxiety compared to female students [45].

Physical activity (PA) is strongly related to mental [46,47]; therefore, it has been highly recommended during COVID-19 by the World Health Organization [48]. During the pandemic, decreased PA intensity has been noted in countries of Asia, America, Africa, and Europe [3,49,50,51], although in some countries (such as Belgium and Canada) [52,53] the intensity has increased. As research shows that PA plays an important role in decreasing stress levels, the intensity of PA during the pandemic should be monitored, particularly in groups vulnerable to mental health problems such as students. 

There are several studies on mental health during the pandemic among Turkish [43], medical [54], nursing [55,56], and midwifery [57] students, although less attention is given to non-medical students’ mental health.

Considering the novelty of the coronavirus pandemic, it is still unclear how factors related to COVID-19 (perception of its impact on students’ well-being), sociodemographic variables, physical activity, and mental health (anxiety and depression) are associated with perceived stress levels in students. Therefore, the aims of this study are to reveal the prevalence of mental health and perceived stress levels and to explore perceived stress predictors in Turkish university students. Considering previous research showing the role of sociodemographic variables such as place of residence and gender, year of study, physical activity, satisfaction with life, and mental health indicators such as anxiety and depression symptoms, we examined those variables as predictors of perceived stress among Turkish students during the COVID-19 pandemic.

## 2. Materials and Methods 

### 2.1. The Study

A cross-sectional study was conducted online between 16 May and 10 June 2020. The first confirmed case of COVID-19 in Turkey was reported on 10 March 2020, and the first COVID-19-related death in the country was announced on 17 March. The number of daily coronavirus cases was 5000–5100 per day, reaching a peak between 11–21 April. From the beginning of June the number decreased to 1000–1500 cases per day and stayed stable until October 2020 [58]. After the number of infections increased, remote learning for the 2020 spring semester was introduced at universities [59]. During the research period, particular measures were adopted by the government in order to prevent the spread of COVID-19 (closing schools and universities; enforcing social distancing; closing non-essential shops, restaurants, gyms, sport facilities, and theaters; and limiting traveling and socializing). 

The survey was created by means of Google Forms. The invitation to the online study was sent to students by researchers via email and social media. For 95% of the sample, academics at three main universities sent invitations to students (Bingol University, 42%; Atatürk University, 43%; Muğla Sıtkı Koçman University, 10%). Within this sample across the three main universities, the invitation was send by academics to about 4500, 410, and 150 students, respectively. The remaining 5% of the sample was collected by both, academics and students, and the invitation was shared with 8000 students; nevertheless, only 19 students responded. 

The survey was usually completed within 20 min. The study adhered to ethical requirements pertaining to the anonymity and voluntariness of participation. An informed consent question was added to the survey, and each participant consented to participate in the study. Initially, 361 persons responded to the invitation. However, ultimately, three respondents did not agree to participate in the study. 

### 2.2. The Sample

Sampling was purposive. The selection criterion was being a Turkish student. To assess the sample size, we used the sampling technique presented by [60]. The calculation, based on a previous study (6.3% prevalence [30], type 1 error of 3%, and study power of 95%), showed that at least 252 individuals would need to be surveyed. Nevertheless, we included 358 Turkish university students due to a high number of variables and the possibility of within-group comparisons. 

The study encompassed 358 students from 13 Turkish universities located in 10 cities: Bingol University, Bingöl (n = 151, 42%); Atatürk University, Erzurum (n = 153, 43%); Muğla Sıtkı Koçman University, Muğla (n = 35, 10%); Başkent University, (n = 3, 0.8%), Beykoz University (n = 1, 0.3%), Boğaziçi University (n = 1, 0.3%), and Istanbul University (n = 1, 0.3%), İstanbul; Ağrı İbrahim Çeçen University, Ağrı (n = 5, 1.4%); Adnan Menderes University, Aydın (n = 1, 0.3%); Fırat University, Elazığ (n = 3, 0.8%); Akdeniz University, Antalya (n = 1, 0.3%); Dicle University, Diyarbakır (n = 1, 0.3%); and Kırıkkale University, Kırıkkale (n = 1, 0.3%). Predominantly, students represented the bachelor level of studies, and the most frequent fields of study represented were social sciences and health sciences. Nearly 70% of participants lived in a city.

Women constituted 56% of the sample (n = 200). The participants’ age ranged from 19 to 40 years, and the average age was 23. Detailed descriptive statistics on gender, place of residence, level of study, field of study, year of study, and type of study are presented in Table 1.

### 2.3. Measures

To measure whether the respondents’ appraised the situation in their life as stressful, the Perceived Stress Scale (PSS-10) [16] in the Turkish adaptation [61] was employed. The PPS-10 consists of 10 items referring to the frequency of stressful events that occurred in the month preceding the study, which are assessed on a 5-point scale (0 = never to 4 = very often). The internal reliability of the Turkish version was adequate at 0.70 [16]. The Cronbach’s α for this sample was 0.75.

The 7-item generalized anxiety disorder (GAD-7) scale [42] in the Turkish adaptation [62] is a self-reported measure designed to screen for symptoms following Diagnostic and Statistical Manual of Mental Disorders, fourth edition (DSM-IV) criteria. People rate how often they experienced anxiety symptoms in the two weeks preceding the study on a 4-point Likert scale (0 = not at all, 1 = several days, 2 = more than half the days, and 3 = nearly every day). The ranges of GAD-7 scores are 0–4, minimal anxiety; 5–9, mild anxiety; 10–14, moderate anxiety; and 15–21, severe anxiety [17]. Scores above 10 points indicate an anxiety disorder. In this study, the Cronbach’s α for the GAD-7 equals 0.91.

The Patient Health Questionnaire (PHQ-8) was used to measure depression symptoms among Turkish students. The PHQ-8 consists of eight items, conforming with DSM-IV diagnostic criteria [19]. Participants use a Likert-type response scale ranging from 0 = not at all to 3 = nearly every day. The ranges of PHQ-8 scores are 0–4, normal; 5–9, mild major depressive disorder; 10–14, moderate; 15–19, moderately severe; and 20–24, severe [45]. A cut-off score of 10 or above is recommended to screen for major depressive disorder [45]. The Turkish language version was derived from the Multicultural Mental Health Resource Centre. The internal consistency reliability of the original version measured by Cronbach’s α equals 0.86, and in this study it was 0.89.

### 2.4. Pandemic-Related Measurements

Perception of COVID-19 impact (PCI) on well-being [46] was measured using five statements rated on a 5-item Likert scale (1 = strongly disagree to 5 = definitely agree). PCI measures how much students are afraid that the current situation related to the coronavirus pandemic may negatively affect their life in each of the following areas: (1) completion of the semester and graduation; (2) job search and professional development; (3) financial situation (e.g., subsistence during studies); (4) relationships with loved ones and family; and (5) relationships with colleagues and friends. Next, scores obtained from the five items were summarized. Higher scores indicated more significant coronavirus-related concerns. Internal reliability of the scale was appropriate with Cronbach’s α = 0.72.

Exposure to COVID-19 [46] was measured by eight questions concerning pandemic consequences related to (1) experiencing COVID-19 symptoms; (2) being tested for COVID-19; (3) being hospitalized due to COVID-19; (4) being in a strict 14-day quarantine; (5) COVID-19 infection in family, friends, or relatives; (6) death in the family; (7) losing a job; and (8) declining economic status.

Physical activity (PA) during the coronavirus-related lockdown [46] was assessed using the following question (Q1): “How many days a week did you practice physical exercises or pursue sports activities at home or away from home, at the university, in a club, or at the gym in the previous month?” Participants answered this question on an 8-point scale (0 = not one day to 7 = seven days a week). The subsequent question (Q2) was: “How many minutes a day (on average) did you practice?” Respondents completed the question by stating an average number of minutes of PA per day. The following question (Q3), answered on an analogous 8-point scale to Q1, was: “How many days a week did you do physical exercise or pursue sports activities at home or away from home, at the university, in a club, or at the gym within a month before the general coronavirus quarantine?” The last question (Q4) was: “How many minutes a day (on average) did you practice?” The number of days was multiplied by the number of minutes per day to calculate the PA level during the pandemic (Q1,2) and before the pandemic (Q3,4). 

PA level was divided into two groups: inactive (less than 150 min weekly) and active (150 or more minutes weekly) in relation to World Health Organization (WHO) [47] recommendations. 

## 3. Results

The study used SPSS.25 (IBM, Armonk, NY, USA) software. The analysis encompassed Pearson’s r coefficient correlations, one-way analysis of variance (ANOVA) to evaluate significance of differences, and hierarchical multiple linear regression. In addition, descriptive statistics of the surveyed variables were presented. Although the analysis did not prove a normal distribution of variables (*p* < 0.05), a further analysis of distribution based on skewness and kurtosis coefficients indicated sufficient symmetry and similarity to the Gaussian curve since the absolute values of the obtained coefficients did not exceed 1, which indicates good psychometric properties [63]. It is noteworthy that ANOVA is considered a robust test against the normality assumption and tolerates violations to its normality assumption well.

### 3.1. Descriptive Statistics

The analysis of descriptive statistics showed that perceived stress was high, generalized anxiety disorder (GAD) was mild, and depression symptoms (PHQ) were moderate, while satisfaction with life among Turkish students during the pandemic was low. Even though average anxiety in the sample was mild, over half of the students presented symptoms according to the GAD (52%) and PHQ (62%). One-third of students met neither depression nor anxiety diagnostic criteria, whereas nearly half were characterized by a dual diagnosis (45%). Almost every fifth university student (23%) met the criteria for at least one diagnosis (anxiety or depression).

Perception of the negative impact of COVID-19 on students’ well-being was the most pronounced regarding their financial situation, job search, and completion of the semester. The least negative impact was perceived in the area of relationships with loved ones, family, and friends. When analyzing results for the perception of the impact of COVID-19 on well-being as a five-item scale, it can be noted that the grand mean was 3.75. It can be interpreted that, in general, students perceived the impact of the pandemic as negative. Before the pandemic, students were physically active for 189 min weekly, and during the pandemic for 75 min weekly. Table 2 outlines the descriptive statistics of the surveyed variables for all scales as sums of the points of all items. 

Referring to the variables related to the COVID-19 pandemic, the percentages of physical activity (PA) before and during the pandemic and exposure to the virus were analyzed. Before the pandemic, 38% of students were physically active and met the WHO criterion (over 150 min per week), while during the pandemic only 13% remained active. The effect of exposure to COVID-19 was the strongest in the aspect of declining economic status (65%) and experiencing job loss (by a student or in the student’s family; 47%). Although 6% of students experienced symptoms of coronavirus infection, only one student was hospitalized. Of the 12% of students tested for COVID-19, 5% were in a strict 14-day quarantine. Every fifth student (22%) experienced COVID-19 infection among close relatives. There were 19 cases of coronavirus-related deaths among students’ families.

Referring to psychological variables, only 13% of students were characterized by a normal level of generalized anxiety disorder (GAD), whereas over half of them (52%) presented clinical symptoms of GAD. Low satisfaction with life was perceived by 56% of students, and high satisfaction by only 19%. A majority of students (71%) reported high levels of perceived stress, while only 6% reported low levels. The results are presented in Table 3. 

### 3.2. Correlations

Pearson’s r coefficient was applied in order to verify correlations among variables (Table 4). The correlations between perceived stress and other variables turned out to be significant. The effect size of the correlation between perceived stress (PSS) and the perception of COVID-19 impact (PCI) on students’ well-being was positive and small, whereas for PSS and satisfaction with life it was negative and small. The correlation between PSS and generalized anxiety disorder intensity can be described as positive and large. PCI was negatively correlated with satisfaction with life and positively with GAD. Both effect sizes were small. Satisfaction with life was inversely correlated with GAD. The effect size was also small. 

### 3.3. Significance of Differences

One-way ANOVA was introduced to assess the significance of differences in perceived stress in relation to gender, place of residence, year of study, and physical activity (PA) during the pandemic. Place of residence and year of study turned out to be insignificant (*p* > 0.005), whereas gender and PA were significant. For statistical purposes, six participants who did not want to describe themselves as male or female were excluded from the analysis. Gender turned out to significantly diversify PSS: F(1350) = 15.79, *p* < 0.001. Female students were characterized by significantly higher PSS levels (M = 23.88, SD = 6.56) compared to male students (M = 21.26, SD = 5.48), although the effect size was small; Cohen’s d = 0.43. PA significantly diversified PSS: F(1356) = 4.04, *p* = 0.045. Inactive students had significantly higher PSS levels (M = 23.16, SD = 6.32) compared to active students (M = 20.00, SD = 5.16). The effect size was medium; Cohen’s d = 0.55.

### 3.4. Hierarchical Multiple Linear Regression

In order to develop a model for predicting perceived stress levels among students during the COVID-19 pandemic, a hierarchical multiple linear regression analysis was conducted. The dependent variables introduced into the regression model were the following: gender, physical activity during the pandemic, perception of negative impact of COVID-19 on well-being, satisfaction with life, and anxiety. The introduced variables were significantly diversified and correlated with perceived stress. Nevertheless, place of residence and year of study turned out to be insignificant for perceived stress differentiation and were therefore excluded from the regression model. An examination of correlations (see Table 4) revealed that no independent variables were highly correlated, with the exception of PSS and GAD-7. However, as the collinearity statistics (tolerance and variance inflation factor VIF) were within accepted limits, the assumption of multicollinearity was deemed to have been met [64,65].

For the first block analysis, the predictor variable gender was analyzed. The results of the hierarchical analysis of model 1 turned out to be statistically significant: F(1350) = 15.79, *p* < 0.001. Being a female student was a predictor of perceived stress and explained only 4% of the variance. For the second block analysis, the predictor variable physical activity during COVID-19 was introduced. Model 2 turned out to be significant (F(2349) = 13.85, *p* < 0.001). Being a physically inactive female student explained 7% of the variance. The perception of a negative impact of COVID-19 on students’ well-being was introduced to model 3. The additional variable positively predicted PSS. Model 3 was statistically significant (F(3348) = 13.28, *p* < 0.001) and explained 9% of the variance. Psychological variables satisfaction with life and generalized anxiety disorder were introduced as predictors to model 4, which also turned out to be significant (F(5346) = 33.81, *p* < 0.001) and explained 32% of the variance. Nevertheless, only two predictors remained statistically significant; high generalized anxiety disorder and being physically inactive during the pandemic explained 32% of the variance. Therefore, after controlling for gender and the perception of a negative impact of COVID-19, mostly generalized anxiety disorder, along with physical inactivity, predicted high perceived stress. The results are shown in detail in Table 5. 

## 4. Discussion

Our study evidenced that students reported high perceived stress, mild generalized anxiety, and low satisfaction with life. More than half of the students met the diagnostic criteria of generalized anxiety disorder (52%) and depression (63%). Female and physically inactive students had higher perceived stress levels. A hierarchical linear regression model showed that after controlling for gender and negative COVID-19 impact on well-being, anxiety and physical inactivity significantly predicted high perceived stress. The study shows that students’ mental health during the pandemic is at high risk. 

### 4.1. Prevalence of Perceived Stress and Mental Health 

In this study, students were characterized in general by high perceived stress, mild anxiety, and moderate depression, with 5.6% reporting low perceived stress, 23% medium, and 71.2% high. The results differed among Indian students, i.e., 4%, 82.7%, and 13.3%, respectively [66]; by comparison, a greater number of Turkish students had higher stress levels. This was similar to results for Polish [29] and Saudi Arabian [42] students. Furthermore, stress levels among Turkish students compared with Swiss students [11], who experienced an increase in perceived stress from low to medium due to the pandemic, were significantly higher. The differences in stress levels can be explained by financial concerns and insufficient financial support in Turkey. It is worth noting that students’ satisfaction with life in Turkey was low. This also draws attention to socioeconomic status as an important predictor of psychological functioning in highly stressful situations such as a pandemic [9]. Moreover, in the Chinese population, satisfaction with life was low before and during the pandemic, and decreased during the pandemic [4,27].

Compared to stress levels of Turkish students before the pandemic [35], current research shows significantly higher levels, with 39.9% of students and 63% of medical students reporting medium and high levels [67] pre-pandemic and 94.4% during the pandemic in our research. The stress level among Turkish nursing students surveyed in April–May 2020 was moderated [55] and therefore different from the high level among students in our study in May–June 2020. The increased stress level in Turkish students is in agreement with other research [39,40].

Turkish students had mild anxiety and moderate depression symptoms. Among those with anxiety, 35.7% reported mild symptoms, 28.8% moderate, and 22.9% severe; 12.6% of students had no anxiety symptoms. Among those with depression, 27.7% reported mild symptoms, 24% moderate, 23.7% moderately severe, and 15.1% severe symptoms; 9.5% of students had no depression symptoms. Depression symptoms were more frequent than anxiety symptoms (62.8% vs. 51.6%). One-third of students (31%) did not meet any diagnostic criteria for anxiety or depression, but almost half (45.5%) had a dual diagnosis. Therefore, the prevalence of comorbid depression encompasses a large number of Turkish university students, although there were more students with severe anxiety (23%) than severe depression (15%). 

Previous research evidenced higher anxiety in students during the pandemic [11], although similar to our sample, at a mild level. Nevertheless, the number of students reporting diagnostic symptoms of generalized anxiety disorder was high (51.6%). On the one hand, such high levels of anxiety in students are in agreement with research showing higher anxiety rates compared to the general population [27,29,44,68]. On the other hand, such levels highly exceed the prevalence of anxiety assessed by the same measurement (GAD-7) in Chinese (4% [44] and 28% [69]), Polish (35%) [29], and Bangladeshi (42.9%) students [45], but were slightly lower compared to Israeli nursing students (56%) [68]. Compared to the Italian general population, the prevalence of severe anxiety symptoms in Turkish students was slightly higher, at 22.91%, compared to 20.8% [32].

Depression levels were moderate in Turkish students, whereas severe depression symptoms were reported by 15%, which is in agreement with a Chinese student sample (16%) [3]. On the other hand, in Wuhan (China), depression symptoms characterized almost half of the student sample (48%). The prevalence of depression symptoms in this research (62.8%) was higher than that in the Bangladeshi student sample (53.4%) [45]. The prevalence of moderately severe to severe depression symptoms in Turkish students was twice as high (38.8%) when compared to the general population (17.3%) [32].

These differences might be due to the different measurement periods during the COVID-19 pandemic. Therefore, although high anxiety and depression are related to the pandemic in general, cross-cultural differences should be taken into account when explaining perceived stress in student populations. On the other hand, the results are in agreement with other research revealing higher stress, anxiety, and depression in students compared to the general population [34].

### 4.2. Exposure to COVID-19 

Direct exposure to COVID-19 was relatively low among Turkish students in terms of hospitalization (only one student was hospitalized). At the same time, every fifth student experienced COVID-19 infection among family or friends, and 5.3% experienced the death of close relatives. By the end of data collection (10 June) there were 4747 deaths due to COVID-19, and while the curve of new cases was decreasing, the number of total deaths was rising. Therefore, it would seem that the student sample was overexposed to the bereavement experience. Nevertheless, there were concerns regarding the openness and clarity of COVID-19 data in Turkey, as it seemed that the prevalence of the disease (particularly total deaths) might be underreported by governments of other countries [70,71].

Turkish students were most affected by the COVID-19 pandemic in the financial aspect. Among the students, 65% noted a deterioration of their economic status, and half of them experienced losing a job (themselves or among family members). This means that the coronavirus pandemic affects healthy people mostly by way of economic consequences. The beginning of the pandemic brought high uncertainty among students regarding academic life (mode of study, completion of studies, or graduation) [72].

Even though such problems were related to the abrupt introduction of remote learning and the uncertainty arising at universities seemed to be the strongest source of stress in students [73], it turned out that financial issues had a stronger impact. In our study, undergraduates perceived a medium negative impact of the pandemic on their university achievements, but a high impact on their financial situation. Therefore, based on our research, the consequences of this unique worldwide situation for students should be labeled as a financial pandemic. 

### 4.3. Exploring Perceived Stress: The Role of Gender, Physical Activity, Perception of COVID-19, Satisfaction with Life, and Anxiety 

We have shown that perceived stress relates positively with a negative perception of COVID-19 impact (PCI) on students’ well-being in the areas of education, economy, and relationships. Perceived stress was strongly linked to generalized anxiety disorder and negatively with satisfaction with life. Place of residence and year of study turned out to be insignificant for stress level differentiation. Previous research showed an inconsistent but significant association between living in a rural or urban area and mental health [44,45] but not perceived stress. Our research shows that place of residence was not associated with perceived stress during the pandemic. We also expected that the year of study, as related to age, would be important for perceived stress, but our hypothesis failed. 

Our findings confirm the importance of physical activity for decreasing perceived stress levels [27]. The students’ physical activity dropped compared to the period before the pandemic, which is also consistent with other studies [3,50]. There was a 25% decrease in physical activity (≤150 min weekly). Inactive students had also significantly higher perceived stress levels compared to active students, and the effect size was medium.

The study revealed significant differences in stress levels due to gender. In relation to other studies, female students reported significantly higher stress levels compared to male students [3,27,29,31,51,74,75,76], but the effect size was small. 

We have shown that being female, being physically inactive, and having a negative perception of the impact of COVID-19 on students’ well-being are significant predictors of perceived stress in students. However, after introducing satisfaction with life and anxiety to the model, gender turned out to be an insignificant predictor. Additionally, satisfaction with life turned out to be insignificant with regard to stress levels during the pandemic. It might be due to the fact that the Satisfaction with Life Scale (SWLS) is a generalized cognitive assessment of subjective well-being that is quite stable throughout the course of life [77,78]. Therefore, as perceived stress was elevated due to the recent experience of the pandemic (as compared to the pre-pandemic period), SWLS could not predict its levels, due to high stability over time. Finally, 32% of the variance was explained by physical inactivity, and high anxiety was the strongest predictor. Hence, high anxiety and physical inactivity are better predictors of perceived stress than gender, even though gender diversifies stress levels. 

The perception of the COVID-19 impact (PCI) on students’ well-being turned out to be a significant predictor of perceived stress in the model of regression together with gender and physical activity. In this model (without anxiety and satisfaction with life), negative PCI, female gender, and physical inactivity predicted high stress levels in students. The PCI components were related to problems with graduation and completion of the semester, job search, financial situation, and concerns about relationships with family and friends. Therefore, unless anxiety is disregarded, a negative perception of the impact of COVID-19 in those areas of students’ lives strongly predicts higher stress levels.

The pandemic situation affects students’ well-being, as the results of a negative PCI (including concerns about academic future and completing the semester) show. Considering a strong relationship between perceived stress, mental health, and problems with academic life with burnout syndrome [79,80,81,82], we want to shed light on possible outcomes of high stress levels that could lead to academic burnout in the future.

### 4.4. Future Research

Future research should focus on comparing student populations to other young adults and adults. The second stage of this study will be carried out in October–November 2020 to determine the longitudinal effects of COVID-19 on mental health and well-being. As most of the pandemic research relates to negative consequences, an analysis of positive outcomes will be introduced in the second wave of the study. 

### 4.5. Limitations

There are several limitations to this study. One is its cross-sectional character. A longitudinal study design would have enabled establishing the impact of the COVID-19 pandemic on perceived stress and the direct cause-and-effect relationship between examined variables and perceived stress. Also, the lack of random sampling and the representation of a student population limited to eastern Turkey seem to be a burden in generalizing the results. Using globally validated standardized tools allowed for discussion with regard to perceived stress and mental health. Nevertheless, the discussion and literature review showed great differences in student populations between countries, and an international study design could help to explain those differences and understand students’ perceived stress levels during the COVID-19 pandemic. In addition, we did not control the number of students with a previous clinical diagnosis of generalized anxiety disorder. Controlling this variable could prove the effect of COVID-19 on anxiety and perceived stress, so as to fully reflect the severity of depression and anxiety symptoms among students.

The biggest limitation to this study is very low response rate (2.7%). The response rates were 3.4% (151/4500) at Bingöl University, 37.3% (153/410) at Atatürk University, 23.3% (35/150) at Muğla Sıtkı Koçman University, and 0.2% (19/8000) at all other 10 universities. The highest response rate at Atatürk University could be due to the form of dissemination of the invitation to the study participation. It was spread directly via the university’s online platform, i.e., a highly formal manner. That could cause a strong feeling of obligation to fulfill the survey by students. The lower response rate in other cases could be due to the use of many communication procedures (i.e., social media) such that they could have missed the survey or were not willing to fill it out. Nevertheless, research with low response rates may generate prevalence estimates that are biased by selective non-response [83]; therefore, the results of our study should be interpreted with caution.

## 5. Conclusions

The state of perceived stress, mental health, and well-being among Turkish students is alarming and requires systemic and dedicated action. Universities together with authorities should collaborate to create an efficient system, of both psychological and financial help for students. Considering the high number of students experiencing a deterioration of economic status, special programs dedicated to financial support during the pandemic should emerge, i.e., scholarships or student loans. Also, fighting an *infodemic* (an overabundance of information) and misinformation during the pandemic can help to improve the mental health of students.

To decrease stress levels, online psychological counseling services should be provided. Even though female students presented higher anxiety and stress levels, decreasing anxiety levels and boosting physical activity for all students seem critical. Providing mental health support systems for students and promoting physical activity on a regular basis could decrease perceived stress levels. 

## Figures and Tables

**Table 1 ijerph-17-08961-t001:** Demographic characteristics of the study sample (N = 358).

Demographic Variables	N	%
Gender		
Women	200	55.87
Men	152	42.46
Did not want to say	6	1.67
Place of residence		
Village	59	16.48
Town	50	13.97
City	249	69.55
Level of study		
Bachelor	282	78.77
Master	48	13.41
Postgraduate	26	7.26
Doctoral	2	0.56
Field of study		
Social science	190	53.07
Natural science	4	1.12
Health science	164	45.81
Year of study		
First	66	18.44
Second	54	15.08
Third	60	16.76
Fourth	157	43.85
Fifth	21	5.87
Type of study		
Stationary	358	100.00

**Table 2 ijerph-17-08961-t002:** Descriptive statistics (n = 358).

				95% CI	
Variable	Range	M	SD	LL	UL
Perceived stress	4–40	22.74	6.26	22.09	23.39
Anxiety	0–21	6.43	4.94	6.19	6.68
Depression	0–24	12.42	5.99	2.94	3.20
Satisfaction with life	5–34	16.72	6.81	16.06	17.49
Perception of impact of COVID-19 on well-being	5–25	18.74	4.57	18.27	19.22
Completion of the semester and graduation	1–5	3.36	1.26	3.30	3.42
Job search and professional development	1–5	3.67	1.20	3.61	3.73
Financial situation	1–5	3.82	1.18	3.76	3.88
Relationships with loved ones, family	1–5	2.44	1.43	2.37	2.51
Relationships with colleagues, friends	1–5	2.57	1.36	2.50	2.64
Physical activity before pandemic (min per week)	0–1890	188.08	289.09	158.03	218.13
Physical activity during pandemic (min per week)	0–1890	75.53	192.31	55.44	95.42

M = mean; SD = standard deviation; CL = confidence interval, LL = lower limit of the confidence interval, UL = upper limit of the confidence interval.

**Table 3 ijerph-17-08961-t003:** Coronavirus-related and psychological variables (generalized anxiety disorder, satisfaction with life, perceived stress) (n = 358). GAD-7, Generalized Anxiety Disorder scale; PHQ-8, Patient Health Questionnaire; SWLS, Satisfaction with Life Scale; PSS-10, Perceived Stress Scale.

Variable	*n*	%
Physical activity ≥ 150 min per week		
Before coronavirus pandemic	136	38.00
During coronavirus pandemic	48	13.41
Exposure to COVID-19		
Symptoms of coronavirus infection	22	6.14
Tested for coronavirus	12	3.35
Hospitalization due to coronavirus	1	0.28
Strict quarantine for at least 14 days	18	5.03
Coronavirus infection in close relatives	79	22.06
Death of close relative due to coronavirus	19	5.31
Job loss because of coronavirus	174	48.60
Deterioration of economic status	232	64.80
Anxiety (GAD-7)		
Normal (0–4)	45	12.57
Mild (5–9)	128	35.75
Moderate (10–14)	103	28.77
Severe (15–21)	82	22.91
Depression (PHQ-8)		
Normal (0–4)	34	9.50
Mild (5–9)	99	27.66
Moderate (10–14)	86	24.02
Moderately severe (15–19)	85	23.74
Severe (20–24)	54	15.08
Neither depression nor anxiety diagnosis (score ≤ 10)	111	31.00
Anxiety only diagnosis (GAD-7 ≥ 10)	185	51.68
Depression only diagnosis (PHQ-8 ≥ 10)	225	62.85
Dual anxiety and depression diagnosis (scores ≥ 10)	163	45.54
Satisfaction with life (SWLS)		
Low (5–17)	202	56.42
Medium (18–23)	88	24.58
High (24–35)	68	18.99
Perceived stress (PSS-10)		
Low (0–13)	20	5.59
Medium (14–19)	83	23.18
High (20–40)	255	71.23

**Table 4 ijerph-17-08961-t004:** Correlation matrix between perceived stress, perceived COVID-19 impact on well-being, satisfaction with life, and general anxiety disorder with Pearson’s r coefficient (n = 358).

Variable	1	2	3	4
Perceived stress	–			
Perceived COVID-19 impact on well-being	0.22 ***	–		
Satisfaction with life	−0.22 ***	−0.14 **	–	
General anxiety disorder	0.55 ***	0.29 ***	−0.24 ***	–

** *p* < 0.01; *** *p* < 0.001.

**Table 5 ijerph-17-08961-t005:** Summary of hierarchical regression analysis of variables predicting perceived stress among students during the COVID-19 pandemic (n = 358).

			95% CI				
Variable	B	SE B	LL	UL	Β	t	*p*
Model 1							
Gender ^a^	−2.62	0.66	−3.51	−1.10	−0.21	−3.97	<0.001
Model 2							
Gender ^a^	−2.61	0.65	−3.52	−1.15	−0.21	−4.02	<0.001
Physical activity ^b^	−3.17	0.94	−5.07	−1.37	−0.17	−3.38	0.001
Model 3							
Gender ^a^	−2.28	0.65	−3.22	−0.85	−0.18	−3.52	<0.001
Physical activity ^b^	−2.76	0.93	−4.63	−0.96	−0.15	−2.97	0.003
Perception of COVID-19 impact	0.24	0.07	0.11	0.38	0.17	3.36	0.001
Model 4							
Gender^a^	−0.86	0.58	−1.75	0.33	−0.07	−1.15	0.136
Physical activity ^b^	−2.06	0.81	−3.68	−0.51	−0.11	−2.54	0.011
Perception of COVID-19 impact	0.06	0.06	−0.03	0.21	0.04	0.92	0.357
Satisfaction with Life	−0.08	0.04	−0.16	0.01	−0.08	−1.85	0.065
Generalized anxiety disorder	0.58	0.06	2.58	3.77	0.49	10.08	<0.001

Note. R^2^ = 0.04 for model 1, *p* < 0.001; R^2^ ∆ = 0.03 for model 2, *p* = 0.001; R^2^ ∆ = 0.03 for model 3, *p* = 0.001; R^2^ ∆ = 0.22 for model 4, *p* < 0.001; total R^2^ = 0.32, *p* < 0.001. ^a^ Gender was coded women = 1, men = 2. ^b^ Physical activity was coded inactive = 0, active = 1.

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
