# Peer review of "Exploring Perceived Stress among Students in Turkey during the COVID-19 Pandemic"

_ijerph, 2020, doi:10.3390/ijerph17238961_

Round 1
Reviewer 1 Report
This paper spends a great deal of time talking about burnout syndrome, but the results have no bearing on burnout syndrome. The hypothesis that high stress may lead to burnout syndrome cannot be investigated using this study. You need a hypothesis which this study is able to test. High stress might lead to burnout. High stress could also lead to heart disease and many other things. That is not what the paper is about.
In the aim, and again in the introduction you said 'our aim is to explain stress levels among students'. Did you mean 'explore stress levels'? You can't explain them. What you actually did was find factors associated with stress, but that doesn't explain stress.
Ref 33 says 'it could be that the youth are most affected by the lockdown'. You cited it and said 'students are most affected by the lockdown', which is overstating it.
The fact you only sampled from Turkish students does not mean you used purposive sampling. It means Turkish students were the target population. A proper explanation of sampling and response is needed to establish whether the results are externally valid. You said 'The invitation to the online study was sent by researchers via email to students'. How many students were emailed? How did you decide which students to email? Could recipients forward the form to other students? Did you ask them to?
You said the study adhered to ethical requirements. Give details of ethical approval obtained. If this wasn't done the paper will have to be rejected.
38% is slightly more than every third student.
19 bereavements from covid-19 among 358 students seems very high. By mid-June there had been about 5000 deaths from covid-19 in Turkey. If each of those deaths had 20 close family members that would mean 0.1% of Turkey's population had a bereavement. In your sample it's 5.3%. It seems at least possible that students who had experienced a bereavement from covid-19 were over-represented, maybe because they were more likely to fill in the form.
Explain how you selected the 5 independent variables for your model.
When you say being female explained variance in 4%, do you mean it explained 4% of variance?
You assume these results are generalisable to the wider student population in Turkey, but since we know nothing about the sampling strategy and these students seem unusually affected by covid-19 we can't necessarily do that.
You are making assumptions about causality, for instance saying low physical inactivity is a cause of stress. It could also be a consequence of stress.
The limitations section doesn't make sense. You seem to be saying that the fact you focused on students rather than trying to study the entire country is a limitation. It's not, it's very sensible. The other limitation is that you did not control the students with previous anxiety - do you mean you didn't exclude them? Fair enough, but in that case some information on the prevalence of anxiety among Turkish students pre-pandemic would be useful.
Author Response
Reviewer1 .
Open Review
(x) I would not like to sign my review report
( ) I would like to sign my review report
English language and style
( ) Extensive editing of English language and style required
(x) Moderate English changes required – the paper was sent to MDPI English pre-edit services
( ) English language and style are fine/minor spell check required
( ) I don't feel qualified to judge about the English language and style
Yes |
Can be improved |
Must be improved |
Not applicable |
|
Does the introduction provide sufficient background and include all relevant references? |
( ) |
(x) |
( ) |
( ) |
Is the research design appropriate? |
(x) |
( ) |
( ) |
( ) |
Are the methods adequately described? |
( ) |
( ) |
(x) |
( ) |
Are the results clearly presented? |
(x) |
( ) |
( ) |
( ) |
Are the conclusions supported by the results? |
( ) |
( ) |
(x) |
( ) |
Comments and Suggestions for Authors
- This paper spends a great deal of time talking about burnout syndrome, but the results have no bearing on burnout syndrome. The hypothesis that high stress may lead to burnout syndrome cannot be investigated using this study. You need a hypothesis which this study is able to test. High stress might lead to burnout. High stress could also lead to heart disease and many other things. That is not what the paper is about. – Yes, we can see your point. We wanted to shed the light on possible relationship of high perceived stress in pandemic and academic burnout syndrome. Incorporating your comment, we have provided changes, and left one paragraph regarding this topic in the discussion part. All references and literature review referring to burnout syndrome, have been excluded from the Introduction part.
- In the aim, and again in the introduction you said 'our aim is to explain stress levels among students'. Did you mean 'explore stress levels'? You can't explain them. What you actually did was find factors associated with stress, but that doesn't explain stress. - We have proposed a new title: Exploring Perceived Stress Among Turkish Students During COVID-19 Pandemic. We have also changed the aim of the study, line 127 “explore stress levels predictors”.
- Ref 33 says 'it could be that the youth are most affected by the lockdown'. You cited it and said 'students are most affected by the lockdown', which is overstating it. – Yes, thank you for this comment. Nevertheless, as we have reorganized the introduction part, this reference was excluded.
- The fact you only sampled from Turkish students does not mean you used purposive sampling. It means Turkish students were the target population. A proper explanation of sampling and response is needed to establish whether the results are externally valid. You said 'The invitation to the online study was sent by researchers via email to students'. How many students were emailed? How did you decide which students to email? Could recipients forward the form to other students? Did you ask them to? – Thank you for this comment. We have elaborated this part,
Lines 136-142: “The survey was created by means of Google Forms. The invitation to the online study was sent by researchers via email and social media to students. The sample consists in 95% of three main university students (Bingol University - 42%, Atatürk University - 43%, Muğla Sıtkı Koçman University – 10%). Within this sample in three main universities, the invitation was send by academics to respectively 4,500, 410, and 150 students. The remaining 5% of sample was collected by both, academics and students, and the invitation was shared within 8,000 students, nevertheless, only 19 students responded.”
lines 150-154: “To asses the sample size, we have used the sampling technique by Pourhoseingholi, Vahedi, & Rahimzadeh (2013). The calculation based on previous study (6.3% prevalence [Wang et al., 2020]; type 1 error of 3% and study power of 95%) showed that at least 252 individuals would need to be surveyed. Nevertheless, we have included 358 Turkish university students due to a high number of variables and possibility of within-group comparisons.
- You said the study adhered to ethical requirements. Give details of ethical approval obtained. If this wasn't done the paper will have to be rejected.- Accepted. lines 468-473: “The study’s protocol was approved by the ethics committee of the University Research Committee at the University of Opole, Poland; decision no. 1/2020. In accordance with the Helsinki declaration, written informed consent was obtained from each student before inclusion. This study is a part of an international research project: Wellbeing of undergraduates during the COVID-19 pandemic: International study, registered at Center for Open Science, OSF: Rogowska, A. M., Kuśnierz, C., Ochnik, D., et al. Well-being of undergraduates during the COVID-19 pandemic. 2020 https://doi.org/10.17605/OSF.IO/BRKGD.”
The number research committee review was given. I will be happy to attach the decision of the university research committee. Additionally, according to the World Health Organization Guidelines on Ethical Issues in Public Health Surveillance, a surveillance study in emergency outbreak situations is clearly exempted from ethical review and oversight (HO guidelines on ethical issues in public health surveillance. Geneva: World Health Organization; 2017. Licence: CC BY-NC-SA 3.0 IGO.). Our online survey was applied in April when the lockdown of Ankara City/Turkey was officially announced.
- 38% is slightly more than every third student. – Yes, you are correct. It was changed to, line 245: 38% of students
- 19 bereavements from covid-19 among 358 students seems very high. By mid-June there had been about 5000 deaths from covid-19 in Turkey. If each of those deaths had 20 close family members that would mean 0.1% of Turkey's population had a bereavement. In your sample it's 5.3%. It seems at least possible that students who had experienced a bereavement from covid-19 were over-represented, maybe because they were more likely to fill in the form. – We can see your point. As other readers may have the same concerns, we have added additional paragraph regarding this issue. Lines 366-374: “Direct exposure to COVID-19 is relatively low among Turkish students in terms of hospitalization (only one student was hospitalized). At the same time, every fifth student experienced COVID-19 infection among family or friends, and 5.3% experienced the death of close relatives. By the end of data collection (10 June) there were 4747 deaths due to COVID-19, and while the curve of new cases was decreasing, the number of total deaths was rising. Therefore, it would seem that the student sample was overexposed to the bereavement experience. Nevertheless, there were concerns regarding the openness and clarity of COVID-19 data in Turkey, as it seemed that the prevalence of the disease (particularly total deaths) might be underreported by governments of other countries [71,72].”
- Explain how you selected the 5 independent variables for your model. We have reorganized the introduction part to show the relevance of mental health (anxiety and depression), satisfaction with life, physical activity, gender, and other sociodemographic variables. We have also added explanation on excluding the year of study and place of residence from the hierarchical regression model. Lines 284-294: “In order to develop a model for predicting perceived stress levels among students during the COVID-19 pandemic, a hierarchical multiple linear regression analysis was conducted. The dependent variables introduced into the regression model were the following: gender, physical activity during the pandemic, perception of negative impact of COVID-19 on well-being, satisfaction with life, and anxiety. The introduced variables significantly diversified and correlated with perceived stress. Nevertheless, place of residence and year of study turned out to be insignificant for perceived stress differentiation, therefore were excluded from the regression model. An examination of correlations (see Table 4) revealed that no independent variables were highly correlated, with the exception of PSS and GAD-7. However, as the collinearity statistics (tolerance and variance inflation factor VIF) were within accepted limits, the assumption of multicollinearity was deemed to have been met [65,66].”
- When you say being female explained variance in 4%, do you mean it explained 4% of variance? – Yes, thank you for noticing this. Line 297: “explained only 4% of variance”
- You assume these results are generalisable to the wider student population in Turkey, but since we know nothing about the sampling strategy and these students seem unusually affected by covid-19 we can't necessarily do that. – We have explained those issues in remark 4 and 7.
- You are making assumptions about causality, for instance saying low physical inactivity is a cause of stress. It could also be a consequence of stress.- Yes, we conform with this remark. It is highly difficult prove the direct cause-and-effect relationship. We have changed the paragraph about physical activity. Lines 105-111: “Physical activity (PA) is strongly related to mental health (e.g., Fibbins et al., 2018; Heijnen et al., 2016), therefore PA was highly recommended during COVID-19 by World Health Organization (2020). During COVID-19 pandemic, the decrease of PA intensity has been noted in in countries of Asia, America, Africa and Europe (Ammar et al., 2020; López-Bueno e al., 2020; Stanton et al., 2020; Wang, Lei et al., 2020)., although in some countries (like Belgium or Canada) (Lesser & Nienhuis, 2020; Constandt et al., 2020) intensity of PA increased. As research show that PA plays an important role in decreasing stress levels, therefore, the intensity of PA during pandemic should be monitored, particularly in groups vulnerable to mental health problems such as students. ”
- The limitations section doesn't make sense. You seem to be saying that the fact you focused on students rather than trying to study the entire country is a limitation. It's not, it's very sensible. The other limitation is that you did not control the students with previous anxiety - do you mean you didn't exclude them? Fair enough, but in that case some information on the prevalence of anxiety among Turkish students pre-pandemic would be useful. – Thank you for those insights. We have changed the Limitation section lines 436-447: “There are several limitations to this study. One of the limitations is its cross-sectional character. Longitudinal design study plan would have enable establishing the impact of COVID-19 pandemic on perceived stress and the direct cause-and-effect relationship between examining variables and perceived stress. Also, the lack of random sampling, and therefore gathering representation of students population limited do eastern Turkey, seems to be a burden in generalizing results. Using globally validated standardized tools allowed as for discussion with regard to perceived stress and mental health. Nevertheless, the discussion and literature review showed great differences between countries in students population, therefore for international study design could help in explaining those differences and to understand perceived stress levels during COVID-19 pandemic in students. In addition, we did not control the number of students with the previous General Anxiety Disorder clinical diagnosis. Controlling this variable could prove the effect of COVID-19 on anxiety and perceived stress, as fully reflect the severity of depressive and anxiety symptoms among students.”

Reviewer 2 Report
The current paper looked the potential effects of covid-19 pandemic on some mental health parameters in a Turkish student sample. Paper would be of interest to the readers of this journal as the topic is very relevant to the current pandemic situation.
Significant improvement of English is needed. Introduction should be re-structured as currently there is not structure, jumping from one topic to another, not cohesive, hard to read and follow their course of thought/reasoning. They have a section on Burnout in introduction, however, they have not measured it in the paper, so I cannot really understand why they have included this section in the introduction. There is no rationale/explanation provided for the type of statistical analyses they have carried out and no explanation included as to why they chose specific variables of interest. Discussion section is very poorly written and most of the findings were not explained thoroughly. There is no measurement or analysis of long-term effects, but the title of the paper also mentions this.
Line 3: Please delete ‘long term’ from the title as you did not collect longitudinal data or run and predictive modelling to be able to talk about the long-term effects in this paper. You may discuss potential long-term effects in the discussion, but this should not be included in the title
Line 36: Please briefly explain psychological distress, high anxiety, and stress levels. How are they similar/different from each other ?
Line 41: Please clarify if this is trait or state anxiety
Line 44: Please re-write ‘It evidences the importance of research within this particular social group’
Lines 44-51: This section ‘Our aim …………… in the future’ should be moved at the end of introduction section as this section details the aim of your study and specific hypothesis.
Lines 50-51: How did you do this ? Predictive modelling ? These analyses are not in the current paper, so please do not mention them.
Line 56: As you collected data in Turkey, it would be good to have figures from Turkey here.
Lines 58-59: Please make it very clear that you are referring to not wellbeing per se but wellbeing in Covid-19 context.
Line 71: Instead of making a list of references at the end of the sentence, could you please add the reference after you have given the relevant evidence. It would be easier for the readers to track which reference they might be interested in.
Lines 71-76: Please add relevant citations.
Line 77: Instead of making a list of references at the end of the sentence, could you please add the reference after you have given the relevant evidence. It would be easier for the readers to track which reference they might be interested in.
Linea 89-101: You were talking about students and how they were affected by the lockdown, then you switched back again to the affects of lockdown on mental wellbeing. There is not cohesion. Yet again in line 101, you are again talking about students.
Lines 93-95: Citation(s) missing.
Line 96: What outcomes, please clarify.
Lines 96-97: Citation(s) missing.
Line 99: These are physical symptoms of anxiety, not types of anxiety, please correct it.
Lines 109-114: Citation(s) missing
Line 119: Burnout syndrome in students ?
Lines 122-126: Citation(s) missing
Lines 127- 129: In a student cohort I assume ?
Line 144: where is this review ?
Lines 144-147: This section should be in the discussion.
Section 2.3: Please explain why you chose to use these specific questionnaires.
Lines 212 and 217: Exposure to COVID-19 and Physical activity measures are not validated I am assuming ? Please clarify.
Lines 233-234: Significance of differences ? t-test, ANOVA…etc. which ones did you use please specify.
Lines 291-292: In the introduction, you should be detailing as to why you chose to look at the effects of gender, place of residence, year of study, and physical activity on perceived stress levels. And why only perceived stress levels but not another measured you used ? There is not introduction/clarification regarding this point.
Lines 303-306: Please explain the reason(s) for including these variables as dependent variables in your model.
Line 331: Could you please start your discussion with summarising your findings ?
Discussion section: Discussion section is very poorly written in general and it was very hard to follow authors train of thought. They did not explain as to why they observed and did not observe specific findings. For instance, in their model 4, they should be explaining why only GAD and PA were significant. Every finding and lack of findings/significance should be discussed thoroughly.
Line 538 : Reference should be written correctly.
Author Response
Reviewer 2
Open Review
(x) I would not like to sign my review report
( ) I would like to sign my review report
English language and style
(x) Extensive editing of English language and style required Accepted. The paper was sent to MDPI English pre-edit services
Yes |
Can be improved |
Must be improved |
Not applicable |
|
Does the introduction provide sufficient background and include all relevant references? |
( ) |
( ) |
(x) |
( ) |
Is the research design appropriate? |
( ) |
(x) |
( ) |
( ) |
Are the methods adequately described? |
( ) |
( ) |
(x) |
( ) |
Are the results clearly presented? |
( ) |
( ) |
(x) |
( ) |
Are the conclusions supported by the results? |
( ) |
( ) |
(x) |
( ) |
Comments and Suggestions for Authors
- The current paper looked the potential effects of covid-19 pandemic on some mental health parameters in a Turkish student sample. Paper would be of interest to the readers of this journal as the topic is very relevant to the current pandemic situation. Thank you for this comment.
- Significant improvement of English is needed. The paper was sent to MDPI English pre-edit services
- Introduction should be re-structured as currently there is not structure, jumping from one topic to another, not cohesive, hard to read and follow their course of thought/reasoning. They have a section on Burnout in introduction, however, they have not measured it in the paper, so I cannot really understand why they have included this section in the introduction. Considering numerous remarks to the Introduction section, we have adopted your suggestion and provided significant changes to the Introduction. Lines: 45-123. We have deleted part regarding burnout syndrome and rewritten, therefore also restructured this section. The present structure includes also other suggestions: (1) general definition and relations between perceived stress and mental health (anxiety and depression), (2)general public mental concerns during COVID-19 pandemic, (3)perceived stress and associated factors during COVID-19 pandemic in general population, (4)vulnerability of students population to mental health issues in pre-pandemic period, with research within Turkish students comparing perceived stress before and during pandemic, (5) the prevalence of stress, anxiety or anxiety, and related factor in student population during pandemic(gender & place of residence), (6) relations between physical activity and mental health, (6) short review of research on mental health among Turkish students during pandemic, (7)rationale of the study.
- There is no rationale/explanation provided for the type of statistical analyses they have carried out and no explanation included as to why they chose specific variables of interest. As we have restructured Introduction part, we have shown the factors related to perceived stress during pandemic and based on that we have built hierarchical regression model. Introduction, Lines: 115-123: “Considering the novelty of the coronavirus pandemic, it is still unclear how factors related to COVID-19 (perception of its impact on students’ well-being), sociodemographic variables, physical activity, and mental health (anxiety and depression) are associated with perceived stress levels in students. Therefore, the aim of this study is to reveal the prevalence of mental health and perceived stress levels and explore perceived stress predictors in Turkish university students. Considering previous research showing the role of sociodemographic variables like place of residence and gender, year of study, physical activity, satisfaction with life, and mental health indicators like anxiety and depression symptoms, we examined those variables as predictors of perceived stress among Turkish students during the COVID-19 pandemic.”
- Discussion section is very poorly written and most of the findings were not explained thoroughly. There is no measurement or analysis of long-term effects, but the title of the paper also mentions this. Line 3: Please delete ‘long term’ from the title as you did not collect longitudinal data or run and predictive modelling to be able to talk about the long-term effects in this paper. You may discuss potential long-term effects in the discussion, but this should not be included in the title– Thank you for this comment. We have already change the title to“Exploring Perceived Stress Among Turkish Students During COVID-19 Pandemic” and add short paragraph regarding possible academic burnout syndrome to the discussion.
- Line 36: Please briefly explain psychological distress, high anxiety, and stress levels. How are they similar/different from each other ? We have restructure the introduction section and add this paragraph. Lines 45-58: “Perceived Stress and Mental Health during COVID-19 Pandemic Psychological stress emerges from imbalance between individual’s perception and external demands (Lazarus and Folkman, 1984). Perceived stress refers to assessment of the degree to which situation in one’s life are appraised as stressful, therefore it is related to subjective assessment of life event (Lazarus, Folkman, 1984). It relates to assessment how unpredictable, uncontrollable, and overloaded individuals find their lives (Cohen, 1983). Mental health is analyzed in this study throughout Generalized Anxiety Disorder (GAD) and depressions symptoms according to DSM IV diagnostic criteria. Based on those criteria GAD is characterized by persistent and excessive worry about a number of different things. It relates to anxiety as a state (Spizer et al., 2006). Depression is one the most common yet treatable mental health disorder (Kroenke and Spitzer, 2002). It refers to a number of symptoms including depressed mood, loss of interests in most or all activities, loss of energy or feeling of worthless (Kroenke et al., 2009). Perceived stress is strongly related to both, anxiety (Mills et al., 2014) and depression symptoms (Salleh, 2008). Nevertheless, the relationship between perceived stress and mental health is complex, and the direct cause-and-effect relationship is not clear (Salleh, 2008).”
- Line 41: Please clarify if this is trait or state anxiety. Yes, thank you. We have added information about anxiety as a state, line 53: “It relates to anxiety as a state”
- Line 44: Please re-write ‘It evidences the importance of research within this particular social group’ Line 42: “Therefore, perceived stress levels and mental health of students during pandemic requires monitoring and in-depth research”.
- Lines 44-51: This section ‘Our aim …………… in the future’ should be moved at the end of introduction section as this section details the aim of your study and specific hypothesis. Yes, thank you. We have rewritten and moved this paragraph to the end of Introduction section. Lines: 118-123.
- Lines 50-51: How did you do this ? Predictive modelling ? These analyses are not in the current paper, so please do not mention them. Yes, we understand our mistake. This remark relates to remarks 3 and 5, and we agree with them. We have included only short paragraph in the discussion section. Lines 424-428: „The pandemic situation affect students’ well-being as results on negative PCI (including concerns about academic future and completing semester) show. Considering a strong relationship between perceived stress, mental health and problems with academic life with the burnout syndrome [80-83], we want to shed a light on possible outcomes of high stress levels that may lead to academic burnout syndrome in the future.”
- Line 56: As you collected data in Turkey, it would be good to have figures from Turkey here. As we have restructured the Introduction section, the body of manuscript has significantly changed, nevertheless, additional information about pandemic situation in Turkey has been added. i.e. Lines 127-130: “The first confirmed case of COVID-19 in Turkey was reported on March 10, 2020 and the first COVID-19-related death in the country was announced on March 17. The number of daily coronavirus cases was 5000-5100 cases per day and reached its peak between 11-21 April,2020. From beginning of June the number has decreased to 1000-1500 cases per day and have stayed stable until October 2020.”
- Lines 58-59: Please make it very clear that you are referring to not wellbeing per se but wellbeing in Covid-19 context. As mentioned before, we have change the Introduction section. Nevertheless, we have incorporated changes according to this suggestion. In the new Introduction (see: answer to remark 3) we tried to be precise about prevalence of variables in pandemic vs pre-pandemic period.
- Line 71: Instead of making a list of references at the end of the sentence, could you please add the reference after you have given the relevant evidence. It would be easier for the readers to track which reference they might be interested in. Yes, thank you. We understand that this is important for clarity of presented material. We have left numerous references only in one case, as it referred to one aspect and was proved in given research examples. Lines: 403-404: “female students reported significantly higher stress levels compared to male students [3,32,57,54,58,19,59,60].”
- Lines 71-76: Please add relevant citations. We have change the whole Introduction part. Please, refer to our answer to your remark 3.
- Line 77: Instead of making a list of references at the end of the sentence, could you please add the reference after you have given the relevant evidence. It would be easier for the readers to track which reference they might be interested in. We agree. Please, refer to our answer to your remark 14.
- Linea 89-101: You were talking about students and how they were affected by the lockdown, then you switched back again to the affects of lockdown on mental wellbeing. There is not cohesion. Yet again in line 101, you are again talking about students. We have change the whole Introduction part. Please, refer to our answer to your remark 3.
- Lines 93-95: Citation(s) missing. We have change the whole Introduction part. Please, refer to our answer to your remark 3.
- Line 96: What outcomes, please clarify. We have change the whole Introduction part. Please, refer to our answer to your remark 3.
- Lines 96-97: Citation(s) missing. We have change the whole Introduction part. Please, refer to our answer to your remark 3.
- Line 99: These are physical symptoms of anxiety, not types of anxiety, please correct it. We have change the whole Introduction part. Please, refer to our answer to your remark 3.
- Lines 109-114: Citation(s) missing We have change the whole Introduction part. Please, refer to our answer to your remark 3.
- Line 119: Burnout syndrome in students ? We referred to academic burnout syndrome. We have change the whole Introduction part. Please, refer to our answer to your remark 3.
- Lines 122-126: Citation(s) missing We have change the whole Introduction part. Please, refer to our answer to your remark 3.
- Lines 127- 129: In a student cohort I assume ? We have change the whole Introduction part. Please, refer to our answer to your remark 3.
- Line 144: where is this review ? We have change the whole Introduction part. Please, refer to our answer to your remark 3.
- Lines 144-147: This section should be in the discussion. Yes, thank you. We have conformed to your suggestion and moved this paragraph to Discussion section. Lines 424-428.
- Section 2.3: Please explain why you chose to use these specific questionnaires. Using short and global measurements enables international comparison of data and is in congruence with participant ethics, as shorter questionnaires are not that psychologically engaging for participants. We have added it in the lines 44-441: “Using globally validated standardized tools allowed as for discussion with regard to perceived stress and mental health.”
- Lines 212 and 217: Exposure to COVID-19 and Physical activity measures are not validated I am assuming ? Please clarify. Exposure to COVID-19 consist of one item factors that are not included for further statistical analysis. We used it for detailed group description. Physical activity intensity was built on WHO recommendations. Those variables have been prepared within the international project (Rogowska et al., 2020, OSF).
- Lines 233-234: Significance of differences ? t-test, ANOVA…etc. which ones did you use please specify. Thank you for noticing this. We have elaborated this sentence. Lines 218-219: “The analysis encompassed Pearson’s r coefficient correlations, the One-Way ANOVA to evaluate significance of differences, and hierarchical multiple linear regression.”
- Lines 291-292: In the introduction, you should be detailing as to why you chose to look at the effects of gender, place of residence, year of study, and physical activity on perceived stress levels. And why only perceived stress levels but not another measured you used ? There is not introduction/clarification regarding this point. In the restructured introduction we show the importance of chosen variables for perceived stress and mental health. We also show the relations between perceived stress to mental health. Additionally, high perceived stress is one of the most noticeable and pronounced changes during pandemic among students.
- Lines 303-306: Please explain the reason(s) for including these variables as dependent variables in your model. Please, see our response to your remark 4.
- Line 331: Could you please start your discussion with summarising your findings ?
Discussion section: Discussion section is very poorly written in general and it was very hard to follow authors train of thought. They did not explain as to why they observed and did not observe specific findings. For instance, in their model 4, they should be explaining why only GAD and PA were significant. Every finding and lack of findings/significance should be discussed thoroughly. Accepted. Yes, thank you this insight. We have elaborated and restructured the Discussion part: (1) Prevalence of perceived stress and mental health, (2) Exposure to COVID-19 (as there were also some issues noticed by Reviewer 1), (Exploring Perceived Stress: the role of Gender, physical activity, perception of the COVID-19, satisfaction with life and anxiety).
- Line 538 : Reference should be written correctly. Yes, thank you: Rogowska, A.M.; Kuśnierz, C.; Ochnik, et al. Project: Well-being of undergraduates during the COVID-19 pandemic. 2020. [CrossRef]

Round 2
Reviewer 1 Report
This paper is greatly improved from the previous version. Many of my concerns turned out to be due to missing explanation rather than actual problems with the study.
I think the biggest limitation of the study is the very low response rate. This needs to be shown clearly. The response rates were 151/4500 (3.4%) at Bingol, 153/410 (37.3%) at Ataturk, 35/150 (23.3%) at MSK and 19/8000 (0.2%) at all other universities. Overall, 2.7%. If you know why response rate was lower at Bingol than the other two main universities, or lower among the 8000 extra students than the rest, please explain.
Low response rate needs to be stated as a limitation in the discussion, with some comment on the implications.
Apart from that I have no remaining concerns.
Author Response
- This paper is greatly improved from the previous version. Many of my concerns turned out to be due to missing explanation rather than actual problems with the study.
Thank you for your engagement in the second review round. We appreciate your positive opinion.
- I think the biggest limitation of the study is the very low response rate. This needs to be shown clearly. The response rates were 151/4500 (3.4%) at Bingol, 153/410 (37.3%) at Ataturk, 35/150 (23.3%) at MSK and 19/8000 (0.2%) at all other universities. Overall, 2.7%. If you know why response rate was lower at Bingol than the other two main universities, or lower among the 8000 extra students than the rest, please explain. - accepted
Low response rate needs to be stated as a limitation in the discussion, with some comment on the implications. - accepted
Thank you for this comment. We have added additional paragraph referring to the aforementioned issue. Lines 652-660:
“The biggest limitation to the this study is very low response rate (2.7%). The response rates were 3.4% (151/4500) at Bingol University , 37.3% (153/410) at Atatürk University, 23.3% (35/150) at Muğla Sıtkı Koçman University and 0.2% (19/8000) at all other 10 universities. The highest response rate at Atatürk University could be due to the form of dissemination the invitation to the study participation. It was spread directly via university’s online platform, thus highly formal manner. That could cause a strong feeling of obligation to fulfill the survey in students. The lower response rate in other cases could be due to use of many communication procedures (i.e. social media) that they could have missed the survey or not willing to fill the survey. Nevertheless, research with low response rates may generate prevalence estimates that are biased by selective non-response [84], therefore the results of our study should be interpreted with carefulness.”
- Apart from that I have no remaining concerns
Thank you for this comment.

Reviewer 2 Report
The paper has improved significantly, thanks for taking my comments/builds into consideration. Just a couple of minor points to add:
- Table 5, please add confidence intervals.
- Section 4. Discussion, please start discussion by summarising your findings.
Author Response
- The paper has improved significantly, thanks for taking my comments/builds into consideration.
Thank you for your comments in the first review round. We do appreciate that you have noticed the difference.
- Just a couple of minor points to add:
- Table 5, please add confidence intervals.
The changes have been introduces to Table 5
- Section 4. Discussion, please start discussion by summarising your findings.
Additional paragraph has been added to Discussion section. Lines 456-462:
“Our study evidenced that students reported high perceived stress, mild generalized anxiety, and low satisfaction with life. More than half of the students met the diagnostic criteria of generalized anxiety disorder (52%) and depression (63%). Female and physically inactive students had higher perceived stress levels. A hierarchical linear regression model showed that after controlling for gender and negative COVID-19 impact on well-being, anxiety and physical inactivity significantly predicted high perceived stress. The study shows that students’ mental health during the pandemic is at high risk. ”
